# A Higher Intake of Energy at Dinner Is Associated with Incident Metabolic Syndrome: A Prospective Cohort Study in Older Adults

**DOI:** 10.3390/nu13093035

**Published:** 2021-08-30

**Authors:** Ygor Hermenegildo-López, Carolina Donat-Vargas, Helena Sandoval-Insausti, Belén Moreno-Franco, Monserrat Rodríguez-Ayala, Jimena Rey-García, José Ramón Banegas, Fernando Rodríguez-Artalejo, Pilar Guallar-Castillón

**Affiliations:** 1Department of Preventive Medicine and Public Health, School of Medicine, Universidad Autónoma de Madrid-IdiPaz, CIBERESP (CIBER of Epidemiology and Public Health), 28029 Madrid, Spain; ygorhl@gmail.com (Y.H.-L.); carolina.donat@imdea.org (C.D.-V.); ms.rodriguezayala@gmail.com (M.R.-A.); jimena.reygarcia@gmail.com (J.R.-G.); joseramon.banegas@uam.es (J.R.B.); fernando.artalejo@uam.es (F.R.-A.); 2Centro de Transfusión de la Comunidad de Madrid, Blood Donation Department, 28032 Madrid, Spain; 3IMDEA-Food Institute, CEI UAM+CSIC, 28049 Madrid, Spain; 4Unit of Nutritional Epidemiology, Institute of Environmental Medicine, Karolinska Institutet, 17177 Stockholm, Sweden; 5Department of Nutrition, Harvard T.H. Chan School of Public Health, Boston, MA 02115, USA; helenagabar@gmail.com; 6Department of Microbiology, Pediatrics, Radiology and Public Health, Universidad de Zaragoza, 50009 Zaragoza, Spain; mbmoreno@posta.unizar.es; 7Fundación Instituto de Investigación Sanitaria de Aragón (IIS Aragón), 50009 Zaragoza, Spain; 8Department of Internal Medicine, Ramón y Cajal University Hospital, 28034 Madrid, Spain

**Keywords:** metabolic syndrome, chronobiology, timing of food, older adults, intake of energy, food intake, dinner intake, eating occasions

## Abstract

A higher energy intake (EI) at night has been associated with a higher risk of obesity, while a higher EI at lunch may protect against weight gain. This study examined the association between EI throughout the day and incident metabolic syndrome (MetS) among older adults. A cohort of 607 individuals aged ≥ 60 free from MetS at baseline was followed from 2008–2010 until 2015. At baseline, habitual EI was assessed on six eating occasions: breakfast, mid-morning snack, lunch, afternoon snack, dinner, and snacking. MetS was defined according to the harmonized definition. Statistical analyses were performed with logistic regression and adjusted for the main confounders, including total EI, diet quality, and physical activity/sedentary behavior. During follow-up, 101 new MetS cases occurred. Compared to the lowest sex-specific quartile of EI at dinner, the OR (95% confidence interval) for incident MetS were: 1.71 (0.85–3.46) in the second, 1.70 (0.81–3.54) in the third, and 2.57 (1.14–5.79) in the fourth quartile (*p*-trend: 0.034). Elevated waist circumference and triglycerides were the MetS components that most contributed to this association. A higher EI at dinner was associated with a higher risk of MetS in older adults. Reducing EI at dinner might be a simple strategy to prevent MetS.

## 1. Introduction

The understanding of circadian rhythm and its impact on health has increased substantially in the last decade. The circadian system is regulated by a central clock located in the suprachiasmatic nucleus and various peripheral tissues located in other brain regions, as well as in the liver, pancreas, gut, white adipose tissue, and skeletal muscle, which act as peripheral clocks. The suprachiasmatic nucleus acts via the autonomic nervous system, and is responsible for the circadian secretion of hormones, as well as the regulation of the temperature throughout the day [1]. Neural and endocrine, as well as behavioral, functions have 24 h rhythms and are major determinants of human metabolism [2].

Some external exposures act as synchronizers of the central and the peripheral clocks. These synchronizers are also known as time-givers (“zeitgebers” in German) [3]. The most relevant of them are the cycles of light and dark exposure [4]. However, there is increasing evidence that nutritional timing, specifically the time of day when energy is consumed, is also part of this regulation [2,5].

The interest in chronobiology has increased from the 1990s due to the identification of relevant genetic determinants of circadian rhythmicity (e.g., the CLOCK gene and its association with obesity) [6,7]. Along with this, the detrimental influence of the misalignment of clocks (such as eating late, shift work, or jet lag) on metabolic risk has contributed further to the emerging body of evidence [8,9].

A higher energy intake (EI) at night is associated with a higher risk of obesity [10], while a higher EI at lunch protects against weight gain [11]. In addition, it has been suggested that avoiding eating late could be a strategy to help prevent obesity and the metabolic syndrome (MetS) [5]. Accordingly, this study aimed to assess, for the first time in the literature, the prospective association between the distribution of EI throughout the day and MetS among community-living older adults in Spain who are at high risk of developing MetS.

## 2. Materials and Methods

### 2.1. Study Design and Participants

Data at baseline were taken from the ENRICA Study, which is a representative sample of the noninstitutionalized Spanish population aged 18 and over, recruited in 2008–2010 [12]. Those aged 60 years or older formed the Seniors-ENRICA-1 cohort which was established at baseline with 3518 participants who were followed up until 2015.

Baseline information was collected in three stages: a telephone interview to obtain data on health status, lifestyle, morbidity, and use of health services; a first home visit, performed by a nurse, to obtain blood samples; and a second home visit by trained staff to obtain information on diet and to perform a physical examination. In 2015, the information was updated for those participants who were still alive and agreed to a new blood collection, constituting a subsample of 1821 older adults. Of these, we excluded 676 individuals with MetS at baseline, 72 with missing values that were unable to estimate MetS at baseline, 453 lacking data to calculate the incidence of MetS in 2015, and 13 with missing values for covariates. Thus, the final analytical sample comprised 607 individuals.

All study participants gave written informed consent. The Clinical Research Ethics Committee of the La Paz University Hospital approved the study.

### 2.2. Dietary Assessment

Food consumption in the previous year was collected using a validated computerized dietary history [13]. This tool allowed for the collection of 860 foods, as well as 184 recipes for dishes commonly eaten in Spain, and includes 127 sets of digitized photographs to better estimate the size of food portions. It automatically converts the food into nutrients and energy using standard Spanish food composition tables [14].

Participants were asked to report all the food consumed and when it was eaten at least once every 15 days. Participants were questioned about food consumed at each eating occasion, as follows: “What do you usually eat for breakfast, lunch, dinner, etc.?”. A total of six eating occasions were considered: breakfast, mid-morning snack, lunch, afternoon snack, dinner, and snacking (food consumed between the previous occasions, including before bedtime and when going out for a drink). To facilitate reporting of the food eaten at lunch and dinner, we asked about the first and second courses, desserts, and beverages consumed, as well as about bread and wine. The mean time to complete the diet history was 54 min.

### 2.3. Metabolic Syndrome

MetS is defined, according to the harmonized definition [15], as having at least three of the following five criteria: abdominal obesity (waist circumference of ≥102 cm in men and ≥88 cm in women); fasting blood glucose of ≥100 mg/dL or receiving antidiabetic drugs; systolic/diastolic blood pressure of ≥130/ 85 mmHg or receiving antihypertensive drugs; serum triglycerides of ≥150 mg/dL; and serum HDL-cholesterol of <40 mg/dL in men and <50 mg/dL in women.

Waist circumference was measured with a flexible inelastic belt-type tape at the midpoint between the last rib and the iliac crest, at the end of a normal exhalation [16]. Blood glucose was determined in 12 h fasting blood samples by the glucose oxidase method [17]. Blood pressure was measured using standard procedures with a validated automatic blood pressure device [18]. Serum triglycerides were measured by the glycerol phosphate oxidase method, and serum HDL-cholesterol by direct elimination/catalase method.

### 2.4. Potential Confounders

At baseline, study participants reported sociodemographic and lifestyle variables, including age, sex, level of education (primary or less, secondary, or university), smoking status (never, former, current), and being an ex-drinker. Physical activity at leisure time and in the household was obtained with the questionnaire used in the EPIC-cohort of Spain and was expressed in metabolic equivalents (MET-h/week) [19]. Participants also reported the number of hours spent watching TV and sleeping (summing up sleeping time at night and during the day). Weight and height were measured using standard procedures, and the body mass index (BMI) was calculated dividing weight by squared height (kg/m^2^). Dieting was evaluated with the question: “Are you on a weight loss diet?”. Diet quality was assessed using the MEDAS index of adherence to the Mediterranean diet. To obtain a high MEDAS index indicates a high adherence to the Mediterranean diet [20]. Finally, participants reported if they had been diagnosed with coronary heart disease, chronic respiratory disease, cancer at any site, osteoarthritis, or arthritis.

### 2.5. Statistical Analysis

Logistic regression models were built to assess the risk of incident MetS. The main independent variables were the percentage of EI at each eating occasion, and the study associations were expressed with odds ratios (OR) and their 95% confidence interval (CI). The percentage of EI was modelled as sex-specific quartiles, and the lowest quartile was the reference group. Two logistic models were built. Model 1 was adjusted for sex, age, education level, and total EI (kcal/day). Model 2 was additionally adjusted for smoking status, ex-drinker status, leisure-time physical activity, physical activity in the household, hours/day spent watching TV, total sleeping time (hours/day), BMI, dieting, ethanol intake (g/day), and MEDAS score, as well as the baseline chronic diagnosed diseases: coronary disease, chronic respiratory disease, cancer at any site, osteoarthritis, or arthritis.

The same type of analysis was performed for each component of the MetS. In addition, an isocaloric substitution model was built to evaluate the effect of replacing the percentage of EI at breakfast by same amount of EI at dinner on the risk of incident MetS [21]. Similar analyses were performed for replacing EI% at lunch or at any other occasion with the same EI% at dinner.

All analyses were performed with STATA software v.13.1 (College Station, TX, USA: StataCorp LP). *p*-Values of <0.05 were considered as statistically significant.

## 3. Results

Among the 607 participants, 312 (51.4%) were women, and mean age was 67.3 years (SD 5.3). The main eating occasions were breakfast, lunch, and dinner, with an EI% of 17.1%, 41.9%, and 27.8%, respectively (Table 1). Compared with participants in the lowest quartile of EI% at breakfast, those in the highest quartile were less frequently current smokers, had less total EI, spent less time watching TV, had a lower BMI, and showed lower ethanol intake. Participants in the highest quartile of EI% at lunch were less educated, with lower total EI, and higher adherence to a Mediterranean diet, and those in the highest quartile of EI% at dinner were more frequently current smokers (Table 2).

At the end of the follow-up period, 101 (16.6%) of the participants developed MetS. Of them, 50 were men and 51 were women. The risk of MetS increased across quartiles of EI% at dinner: the OR (95% CI) of MetS was 1 for the lowest quartile, 1.71 (0.85–3.46) for the second, 1.70 (0.81–3.54) for the third, and 2.57 (1.14–5.79) for the highest quartile (P-trend: 0.034). No associations were found for EI% any other eating occasion (Table 3).

Isocaloric replacement of EI at breakfast with the same amount of EI at dinner was associated with a higher risk of MetS, so that the OR (95% CI) of MetS was 2.73 (1.31–5.68; *p*-trend: 0.011) for the highest vs. lowest quartile of EI at dinner. Likewise, replacement of EI at any other time of the day with the same percentage of EI at dinner was associated with a higher risk of MetS (OR: 2.42; 95% CI: 1.22–4.81; *p*-trend: 0.019) (Table 4).

With regard to the five components of the MetS, we found that participants in the highest vs. lowest quartile of EI at dinner had a higher risk of abdominal obesity (OR 2.15; (95% CI: 1.08–4.25; *p*-trend: 0.013), as well as a significant tendency for elevated triglyceridemia (*p*-trend: 0.025) (Table 5).

## 4. Discussion

In this prospective cohort of older adults, a higher percentage of EI at dinner was associated with an increased risk of MetS. Replacing the EI consumed at breakfast or at any other eating occasion with the same amount of energy at dinner also increased the risk of the MetS. Elevated waist circumference and triglycerides were the MetS components that most contributed to this association. In Spain, a country where dinner is usually eaten later than in other countries (generally after 9:00 p.m. [22]), to eat more energy at this eating occasion was detrimental and associated with the development of the MetS.

Previous studies are in line with these findings. However, the exposures were measured differently, and it is difficult to establish comparisons. The association between meal timing and MetS has been previously studied in Japan and Korea. In a cross-sectional study of Japanese people aged 20–75 years, late-night dining habits were associated with a higher risk of the MetS [23]. In another cross-sectional study of Korean adults aged 19 or older (KNHANES Study), night eating (≥25% of total EI at night) was associated with a higher frequency of MetS among males [24]. Finally, in a prospective study of Japanese adults aged 40–54 years, a positive association between dining immediately before going to bed and the risk of incident MetS did not reach statistical significance; however, women who had both the habits of dining just before going to bed and of snacking after dinner had an increased risk of MetS [25].

Our results are in line with current knowledge on nutritional chronobiology [3,5], as well as with previous findings in observational studies [23]. However, the different ways to assess eating timing in relation to MetS, which mainly depends on data availability, make the studies difficult to compare. For instance, some studies considered only two time bands, daytime and evening eating [9,24], while we considered six eating occasions, including snacking, during each 24 h period. Likewise, in some previous studies, the exposure was an intake of at least 25% of energy after 9:00 p.m. [24], or as having dinner just before going to bed [25].

Regarding MetS components, there is some evidence that chronobiology and timing in EI are associated with obesity [10,26]. Specifically, a higher EI during the night was associated with a higher risk of obesity [26]. Of note is that abdominal obesity is a central component of MetS, and can lead to lipid abnormalities, alterations of glucose metabolism, and elevated blood pressure [27]. However, although energy intake timing seems to be relevant in the development of obesity, some inconsistencies have also been observed concerning energy intake at dinner. A previous study conducted in participants 18 years old and over found that while a higher energy intake at lunch was associated with a lower risk of weight gain, no association was found between the energy consumed at dinner and weight gain (>3 kg after 3 years of follow-up) [11].

Regarding hypertriglyceridemia, an experiment with mice found that a high-fat diet at the end of their active period was associated with hypertriglyceridemia, weight gain, increased body adiposity, decreased glucose tolerance, and hyperleptinemia [28]. In addition, another study with mice that were CLOCK gene mutants (with hyperphagia during their sleep/inactivity phase) showed increased weight gain and hypertriglyceridemia [29]. In humans, timing misalignments (i.e., shift work) have been frequently studied, and an association with hypertriglyceridemia was also found; a study among night-shift nurses showed a higher EI at night as well as higher levels of triglycerides, LDL-cholesterol, and total cholesterol. Night shift was also associated with higher levels of HDL-cholesterol [9].

In our analyses, EI at dinner was not associated with other components of the MetS, such as hyperglycemia, high blood pressure, or low HDL-cholesterol. However, these associations were found in some cross-sectional studies. For example, night-shift nurses showed elevated fasting glucose [9]. In addition, in the KNHANES Study, an association between eating at night and reduced HDL-cholesterol was found among men [24].

Although having a regular breakfast has been suggested to decrease adiposity [30], in our analyses no association was found between EI at breakfast and risk of MetS. However, skipping breakfast, which could influence energy balance throughout the day, was very rare (0.7%) in our sample. Furthermore, our results did not show an association between snacking and MetS, although the association was positive, again not reaching statistical significance. Of note is that snacking varies greatly among different populations [31], and in our sample it accounted for only 5% of total EI.

The increasing body of evidence on the influence of EI throughout the day and its impact on metabolic health makes the association between EI timing and the development of MetS plausible [2]. For example, it has been suggested that insulin (the hormone that is altered in MetS) is involved in the post-prandial synchronization of the circadian clocks throughout the body [32]. Another factor that could be acting is the proximity of food intake to the nocturnal rise in melatonin. Calorie intake closer to this nocturnal rise was associated with increased adiposity and impaired glucose homeostatic [33]. It is also known that circadian clocks influence both appetite and energy expenditure, which are ultimately linked to metabolic disorders [34]. Finally, the present study also contributes to the epidemiological and population-based support for the influence of energy timing on cardiometabolic risk management [31].

### 4.1. Importance and Practical Consequences

Our findings suggest that reducing %EI at dinner may serve to lower the risk of MetS. This is important for a number of reasons. First, because of the high prevalence of MetS syndrome (>40%) in our older population [35]. Second, because MetS has been associated with an increased risk of diabetes [36,37] and is a major contributor to the epidemic of cardiovascular disease in the Western world [38]. Third, because MetS does not easily revert. Fourth, because metabolic conditions associated with misalignments do not seem to have a clear genetic component and are based on unhealthy lifestyles [5]. Fifth, because misalignments are becoming more frequent due to long working days and the use of electronic devices at bedtime before sleeping. Finally, because changes in EI timing might be easier to follow than just caloric restrictions. Similar approaches have been proposed to prevent obesity [39,40].

### 4.2. Strengths and Limitations

The main strength of this study is its longitudinal design. However, although the cohort was initially representative of the older adult population in Spain, attrition bias does not allow us to ensure representativeness. Moreover, analyses were adjusted for a significant number of potential confounders, such as total energy, physical activity during leisure time and in the household, watching TV, sleeping, dieting, and alcohol consumption. In addition, certified personnel performed all measurements following standardized protocols.

The main limitation is that diets were self-reported, so recall bias or social desirability bias cannot be ruled out. In addition, despite each participant reporting the food consumed on each eating occasion, the time of the meals was not collected. A diet history validation was performed for food groups with many nutrients and total energy [13], but a specific validation for eating occasions was not performed. On the other hand, the specific time of the six eating occasions was not recorded, although these are quite characteristic of the dietary habits in the Spanish population. Additionally, the influence of sleep preferences was not considered in the analyses. Finally, we cannot rule out some selection bias due to the exclusion of participants with lacking data.

## 5. Conclusions

In this study of older adults, a higher percentage of EI at dinner was associated with a higher incidence of MetS, mostly due to abdominal obesity and hypertriglyceridemia. These findings suggest that a reduction of energy intake at dinner may be useful to prevent MetS. However, the efficacy of this strategy must be proven in future clinical trials with populations.

## Figures and Tables

**Table 1 nutrients-13-03035-t001:** Energy intake at each eating occasion, and percentage of individuals who skipped each eating occasion, among the study participants at baseline. Seniors-ENRICA-1 cohort study, 2008–2010. *N* = 607.

		Energy Intake (kcal/day)		
Eating Occasions	% of Energy Intake	Mean	SD	Skipping Eating Occasion (%)
Breakfast	17.1	339.2	210.5	0.7
Mid-morning snack	4.2	89.0	151.8	41.5
Lunch	41.9	840.4	284.7	0.0
Afternoon snack	4.0	82.3	118.2	38.4
Dinner	27.8	563.2	235.7	0.3
Snacking	5.0	108.3	165.9	37.7
Total energy	100.0	2022.5	559.4	-

**Table 2 nutrients-13-03035-t002:** Baseline characteristics of the participants in the Seniors-ENRICA-1 cohort study according to sex-specific quartiles of the percentage of energy intake at breakfast, lunch, and dinner. *N* = 607.

	Breakfast †		Lunch †		Dinner †	
	Q1	Q4	Q1	Q4	Q1	Q4
**Sex, % of women**	51.3	51.7	51.3	51.7	51.3	51.7
**Age (years), mean**	67.0 (5.6)	67.6 (6.2)	66.9 (5.9)	67.7 (5.6)	67.4 (5.8)	67.2 (5.6)
	**Breakfast †**		**Lunch †**		**Dinner †**	
Level of education, %						
Primary or less	42.1	41.1	35.5	45.7	39.5	42.4
Secondary	31.6	31.8	30.9	31.8	30.9	31.1
University	26.3	27.2	33.6	22.5 *	29.6	26.5
Smoking status, %						
Never smoker	53.3	66.2	53.3	58.9	62.5	53.0
Former smoker	32.2	27.8	33.6	33.1	30.3	30.5
Current smoker	14.5	6.0 **	13.2	8.0	7.2	16.6 **
Ex-drinker, %	11.2	6.0	12.5	6.0	7.9	6.0
Energy (kcal/day), mean	2095 (562)	1951 ** (591)	2184 (637)	1924 *** (501)	2016 (609)	2001 (516)
Physical activity during leisure time (METs h/week), mean	23.2 (14.6)	25.7 (18.1)	24.1 (17.2)	22.3 (16.6)	23.9 (18.5)	24.4 (15.3)
Physical activity in the household (METs h/week), mean	37.8 (31.2)	35.6 (28.2)	37.4 (30.9)	38.6 (28.8)	38.0 (32.2)	33.8 (28.4)
Watching TV (h/week), mean	18.2 (11.6)	15.4 * (10.4)	15.4 (10.3)	15.9 (11.2)	16.4 (12.6)	17.7 (10.1)
Sleeping time (hour/day), mean	7.2 (1.2)	7.2 (1.3)	7.1 (1.5)	7.3 (1.5)	7.2 (1.4)	7.2 (1.3)
Body mass index (kg/m2), mean	27.5 (3.4)	26.8 * (3.6)	26.7 (3.3)	27.6 (4.0)	26.7 (3.6)	27.2 (3.2)
Dieting, %	10.5	6.0	7.2	9.3	6.6	8.6
Ethanol intake (g/day), mean	15.6 (22.1)	8.6 ** (15.1)	9.8 (17.9)	12.7 (16.7)	11.2 (16.8)	14.6 (22.2)
MEDAS score, mean	7.8 (1.8)	7.4 (1.7)	6.8 (2.0)	7.8 *** (1.7)	7.3 (1.9)	7.6 (1.6)
Coronary disease, %	0	2.0	0.7	0	1.3	0
Chronic respiratory disease, %	9.2	6.6	5.9	8.0	6.6	6.0
Cancer, %	2.6	2.0	2.0	1.3	2.0	2.7
Osteoarthritis, %	33.6	38.4	36.8	41.7	37.5	38.4
Arthritis, %	12.5	7.3	11.2	6.0	7.2	13.9

Q1, quartile 1 (lowest); Q4, quartile 4 (highest); MET, metabolic equivalents; MEDAS, Mediterranean Diet Adherence Screener. * *p* for trend <0.05, ** *p* for trend < 0.01, *** *p* for trend < 0.001. † Cut-off points for the quartiles in men: breakfast Q1: (0–11.0), Q2: (11.0–15.2), Q3: (15.2–21.0), Q4: (21.2–62.1); lunch Q1: (14.8–36.5), Q2: (36.6–43.4), Q3: (43.4–49.6), Q4: (49.7–71.0; dinner Q1: (0–22.4), Q2: (22.5–28.6), Q3: (28.6–34.1), Q4: (34.1–59.8). Cut-off points for the quartiles in women: breakfast Q1: (0.1–11.5), Q2: (11.7–17.0), Q3: (17.1–21.5), Q4: (21.5–67.4); lunch Q1: (11.0–34.0), Q2: (34.1–40.8), Q3: (40.8–46.0), Q4: (46.1–78.4); dinner Q1: (0–21.0), Q2: (21.1–27.4), Q3: (27.5–32.0), Q4: (32.0–53.8).

**Table 3 nutrients-13-03035-t003:** Odds ratios (95% confidence interval) for incident metabolic syndrome (2008/10 to 2015) according to sex-specific quartiles of the percentage of energy at each eating occasion among participants in the Seniors-ENRICA-1 cohort study. *N* = 607.

		Model 1	Model 2
	*N*/Cases	OR (95% CI)	OR (95% CI)
Breakfast †	607/101		
Quartile 1 (lowest)	152/25	Ref	Ref
Quartile 2	152/33	1.51 (0.83–2.73)	1.46 (0.78–2.75)
Quartile 3	152/26	1.12 (0.58–2.17)	1.28 (0.64–2.55)
Quartile 4 (highest)	151/17	0.79 (0.35–1.80)	0.84 (0.35–1.97)
*p* for trend		0.619	0.815
Mid-morning snack †	607/101		
Quartile 1 (lowest)	252/43	Ref	Ref
Quartile 2	119/23	1.32 (0.73–2.41)	1.26 (0.67–2.37)
Quartile 3	118/18	0.98 (0.51–1.86)	0.97 (0.50–1.91)
Quartile 4 (highest)	118/17	1.14 (0.55–2.36)	1.09 (0.51–2.34)
*p* for trend		0.838	0.910
Lunch †	607/101		
Quartile 1 (lowest)	152/22	Ref	Ref
Quartile 2	152/24	0.99 (0.51–1.91)	1.05 (0.53–2.08)
Quartile 3	152/24	0.99 (0.50–1.98)	1.10 (0.53–2.29)
Quartile 4 (highest)	151/31	1.62 (0.73–3.58)	1.71 (0.73–3.97)
*p* for trend		0.300	0.258
Afternoon snack †	607/101		
Quartile 1 (lowest)	233/39	Ref	Ref
Quartile 2	125/25	1.34 (0.74–2.41)	1.31 (0.71–2.43)
Quartile 3	125/19	0.96 (0.51–1.81)	1.12 (0.58–2.17)
Quartile 4 (highest)	124/18	1.07 (0.53–2.15)	1.05 (0.50–2.19)
*p* for trend		0.978	0.871
Dinner †	607/101		
Quartile 1 (lowest)	152/16	Ref	Ref
Quartile 2	152/27	1.76 (0.89–3.46)	1.71 (0.85–3.46)
Quartile 3	152/26	1.69 (0.84–3.41)	1.70 (0.81–3.54)
Quartile 4 (highest)	151/32	2.31 (1.06–5.03) *	2.57 (1.14–5.79) *
*p* for trend		0.054	0.034
Snacking †	607/101		
Quartile 1 (lowest)	229/38	Ref	Ref
Quartile 2	127/22	1.09 (0.60–2.00)	1.17 (0.62–2.19)
Quartile 3	126/24	1.30 (0.71–2.37)	1.15 (0.61–2.16)
Quartile 4 (highest)	125/17	1.00 (0.48–2.10)	1.07 (0.49–2.34)
*p* for trend		0.723	0.748

** p* < 0.05. Model 1 was mutually adjusted for the percentage of energy intake consumed at each eating occasion as appropriate, as well as sex, age, level of education (primary or less, secondary, or university), and total energy intake (kcal/day). Model 2 was adjusted as per model 1, plus smoking status (never, former, and current smokers), ex-drinker status, leisure-time physical activity (METs h/week), physical activity in the household (METs h/week), watching TV (hours/week), sleeping time (hours/day), body mass index (kg/m^2^), dieting, ethanol intake (g(day), Mediterranean Diet Adherence Screener score, coronary diseases, chronic respiratory disease, cancer, osteoarthritis, and arthritis. † Cut-off points for the quartiles in men: breakfast Q1: (0–11.0), Q2: (11.0–15.2), Q3: (15.2–21.0), Q4: (21.2–62.1); mid-morning snack Q1: (0), Q2: (0.1–3.0), Q3: (3.1–8.9), Q4: (9.0–51.8); lunch Q1: (14.8–36.5), Q2: (36.6–43.4), Q3: (43.4–49.6), Q4: (49.7–71.0); afternoon snack Q1: (0), Q2: (0–2.9), Q3: (3.0–6.6), Q4: (6.7–34.9); dinner Q1: (0–22.4), Q2: (22.5–28.6), Q3: (28.6–34.1), Q4: (34.1–59.8); snacking Q1: (0), Q2: (0.2–3.9), Q3: (3.9–8.2), Q4: (8.3–40.2). Cut-off points for the quartiles in women: breakfast Q1: (0.1–11.5), Q2: (11.7–17.0), Q3: (17.1–21.5), Q4: (21.5–67.4); mid-morning snack Q1: (0), Q2: (0.2–3.3), Q3: (3.3–6.8), Q4: (6.9–32.1); lunch Q1: (11.0–34.0), Q2: (34.1–40.8), Q3: (40.8–46.0), Q4:(46.1–78.4); afternoon snack Q1: (0), Q2: (0.3–3.5), Q3: (3.5–7.5), Q4: (7.5–31.3); dinner Q1: (0–21.0), Q2: (21.1–27.4), Q3: (27.5–32.0), Q4: (32.0–53.8); snacking Q1: (0), Q2: (0.2–3.7), Q3: (3.7–9.7), Q4: (9.7–53.8).

**Table 4 nutrients-13-03035-t004:** Odds ratios (95% confidence interval) for incident metabolic syndrome among participants in the Seniors-ENRICA-1 cohort study (2008/10 to 2015) when isocaloric substitution of breakfast, lunch, or all other occasions of energy intake for energy intake at dinner. *N* = 607.

		Model 1	Model 2
	*N*/Events	OR (95% CI)	OR (95% CI)
Isocaloric substitution of energy consumed at breakfast for dinner	607/101		
Quartile 1 (lowest)	152/16	Ref	Ref
Quartile 2	152/27	1.79 (0.92–3.52)	1.76 (0.87–3.55)
Quartile 3	152/26	1.76 (0.89–3.49)	1.75 (0.85–3.57)
Quartile 4 (highest)	151/32	2.53 (1.26–5.07) **	2.73 (1.31–5.68) **
*p* for trend		0.014	0.011
Isocaloric substitution of energy consumed at lunch for dinner			
Quartile 1 (lowest)	152/16	Ref	Ref
Quartile 2	152/27	1.72 (0.87–3.37)	1.70 (0.84–3.43)
Quartile 3	152/26	1.58 (0.80–3.15)	1.57 (0.76–3.22)
Quartile 4 (highest)	151/32	1.97 (0.97–4.01)	2.17 (1.03–4.54) *
*p* for trend		0.095	0.065
Isocaloric substitution of energy consumed at all other occasions for dinner			
Quartile 1 (lowest)	152/16	Ref	Ref
Quartile 2	152/27	1.79 (0.92–3.49)	1.76 (0.88–3.53)
Quartile 3	152/26	1.70 (0.87–3.32)	1.66 (0.82–3.36)
Quartile 4 (highest)	151/32	2.26 (1.18–4.35) *	2.42 (1.22–4.81) *
*p* for trend		0.024	0.019

* *p* < 0.05, ** *p* < 0.01. Models 1 and 2 were adjusted as in Table 3.

**Table 5 nutrients-13-03035-t005:** Odds ratios (95% confidence interval) for the incidence of each component of metabolic syndrome according to sex-specific quartiles of percentage of energy intake at dinner among participants in the Seniors-ENRICA-1 cohort study (2008/10 to 2015).

	Model 1		Model 2	
	*N*/Events	OR (95% CI)	*N*/Events	OR (95% CI)
Abdominal Obesity	647/176		647/176	
Quartile 1 (lowest)	162/36	Ref	162/36	Ref
Quartile 2	162/40	1.18 (0.69–2.02)	162/40	1.10 (0.61–2.00)
Quartile 3	162/48	1.62 (0.93–2.81)	162/48	1.82 (0.98–3.36)
Quartile 4 (highest)	161/52	1.99 (1.06–3.74) *	161/52	2.15 (1.08–4.25) *
*p* for trend		0.020		0.013
Hyperglycemia/diabetes	834/150		834/150	
Quartile 1 (lowest)	209/37	Ref	209/37	Ref
Quartile 2	209/29	0.75 (0.44–1.30)	209/29	0.75 (0.43–1.30)
Quartile 3	209/38	1.04 (0.61–1.77)	209/38	1.00 (0.58–1.73)
Quartile 4 (highest)	207/46	1.40 (0.75–2.59)	207/46	1.34 (0.72–2.50)
*p* for trend		0.239		0.308
Arterial hypertension	241/102		241/102	
Quartile 1 (lowest)	61/28	Ref	61/28	Ref
Quartile 2	60/23	0.81 (0.37–1.74)	60/23	0.79 (0.36–1.76)
Quartile 3	61/23	0.73 (0.33–1.58)	61/23	0.75 (0.33–1.70)
Quartile 4 (highest)	59/28	0.87 (0.36–2.11)	59/28	0.86 (0.33–2.22)
*p* for trend		0.656		0.671
Hypertriglyceridemia	1141/98		1141/98	
Quartile 1 (lowest)	286/24	Ref	286/24	Ref
Quartile 2	285/18	0.80 (0.42–1.52)	285/18	0.82 (0.42–1.60)
Quartile 3	286/28	1.53 (0.82–2.87)	286/28	1.66 (0.87–3.19)
Quartile 4 (highest)	284/28	1.92 (0.93–3.97)	284/28	2.07 (0.98–4.35)
*p* for trend		0.038		0.025
Low HDL-cholesterol	1109/156		1109/156	
Quartile 1 (lowest)	278/36	Ref	278/36	Ref
Quartile 2	277/43	1.20 (0.74–1.96)	277/43	1.18 (0.71 -1.95)
Quartile 3	277/40	1.06 (0.64–1.77)	277/40	1.12 (0.66 –1.92)
Quartile 4 (highest)	277/37	0.86 (0.47–1.57)	277/37	0.89 (0.48–1.67)
*p* for trend		0.626		0.773

* *p* < 0.05. Models 1 and 2 were adjusted as in Table 3.

## Data Availability

The datasets generated during and/or analyzed during the current study are available from the corresponding author on reasonable request.

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
