# Peer review of "A Higher Intake of Energy at Dinner Is Associated with Incident Metabolic Syndrome: A Prospective Cohort Study in Older Adults"

_nutrients, 2021, doi:10.3390/nu13093035_

Round 1

Reviewer 1 Report

This article provides an in-depth analysis of the prospective association between the distribution of energy intake (EI) throughout the day and the metabolic syndrome (MetS) among community-living older adults in Spain. Here the Authors focus on incident MetS  according to sex- specific quartiles of the percentage of EI at each eating occasion among participants in the Seniors-ENRICA-1 cohort study, which are at high risk of developing MetS, and find that higher percentage of EI at dinner was  associated with an increased risk of abdominal obesity, as well as a significant tendency for elevated triglyceridemia.

The objective of the paper is clearly stated in the Introduction. The statistical analysis is appropriate for the way that the study is designed and the data are collected. Results are clearly reported. The discussion of the study is sufficiently interpreted by the Authors. Sometimes references are not updated properly (See also Horn C et al. Br J Nutr. 2021; Muscogiuri G et al, Minerva Med. 2021).

Specific Comments

Overall, I find this article very interesting and well described. The evaluation of the eating architecture appears to emerge as one of the most relevant external cues that regulate circadian clock genes in peripheral tissues. As also the Authors state, there is a growing interest on the meal timing as a determinant of individual EI in people with obesity, particularly the association between higher EI at night and higher risk of obesity  and poor glycaemic control in patients with T2DM.

However, data of the same Authors (Ref # 11) previously evidenced that while higher %EI at lunch was associated with a lower risk of weight gain, no association was found between weight gain and the %EI at the rest of the eating occasions, including dinner. Please could the Authors better argue this apparent discordance with their present results ?

In Materials and Methods, please explain why participants were requested to collect urine samples.

In Materials and Methods, it is not clear whether the dietary interview is assessed by a trained staff or the diet are self-reported, as reported in the limitations.

In the Result section, participants in the highest quartile of EI% at lunch have higher adherence to a Mediterranean diet. Please, comment this results in the Discussion section.

In Discussion section, the putative role of insulin or melatonin as regulatory signals to peripheral circadian clocks remains largely speculative. Please could the Authors better explain the role of these hormones, if any, in influencing their present data?

The Authors state that the main strength of this study is its longitudinal design. However, the number of participants lost is very high (67%). Please, discuss this point in the limitation.

Again, among limitations, the Authors reported that the time of the meals was not collected.  This is of particular relevance in limiting the results of this study considering the emerging evidence on the positive impact of time-restricted feeding on metabolic outcomes (see also Currenti W et al Nutrients. 2021). Please, discuss this point.

In the same context, no information is given on sleep preferences (see also Charlot A et al Nutrients. 2021). Again, comment this limitation.

Author Response

This article provides an in-depth analysis of the prospective association between the distribution of energy intake (EI) throughout the day and the metabolic syndrome (MetS) among community-living older adults in Spain. Here the Authors focus on incident MetS  according to sex- specific quartiles of the percentage of EI at each eating occasion among participants in the Seniors-ENRICA-1 cohort study, which are at high risk of developing MetS, and find that higher percentage of EI at dinner was  associated with an increased risk of abdominal obesity, as well as a significant tendency for elevated triglyceridemia.

The objective of the paper is clearly stated in the Introduction. The statistical analysis is appropriate for the way that the study is designed and the data are collected. Results are clearly reported. The discussion of the study is sufficiently interpreted by the Authors. Sometimes references are not updated properly (See also Horn C et al. Br J Nutr. 2021; Muscogiuri G et al, Minerva Med. 2021).

Authors: Thank you very much for the comments and for giving us the update references. We have adding them as a similar example to address obesity. Please, see line 306:

“Similar approaches have been proposed to prevent obesity. [39, 40]”

Specific Comments

Overall, I find this article very interesting and well described. The evaluation of the eating architecture appears to emerge as one of the most relevant external cues that regulate circadian clock genes in peripheral tissues. As also the Authors state, there is a growing interest on the meal timing as a determinant of individual EI in people with obesity, particularly the association between higher EI at night and higher risk of obesity  and poor glycaemic control in patients with T2DM.

However, data of the same Authors (Ref # 11) previously evidenced that while higher %EI at lunch was associated with a lower risk of weight gain, no association was found between weight gain and the %EI at the rest of the eating occasions, including dinner. Please could the Authors better argue this apparent discordance with their present results ?

Authors: The reviewer is right in pointing out this inconsistency. We found 3 main differences between the two studies (reference #11 and the present manuscript). In the first reference the mean age was 53 years (18 and over), while in the present study it is 67 years (60 and over). The second difference is the end-point, while in the first study it was the increase in 3 kilograms over 3 years of follow-up, in the present manuscript it is the development of abdominal obesity after 6 years of follow-up. Finally, in study the #11 we did not exclude participants based on their previous weight, however in the present study those with prevalent abdominal obesity have been excluded. Consequently, the main association is observed among those who reach 60 years of age without having previously suffered from abdominal obesity.

We acknowledge this inconsistency in the text . We have adding the following (line 258):

“However, although energy intake timing seems to be relevant in the development of obesity, some inconsistencies have also been observed concerning dinner. A previous study conducted in participants 18 years old and over, found that while a higher energy intake at lunch was associated with a lower risk of weight gain, no association was found between the energy consumed at dinner and weight gain (>3 kilograms after 3 years of follow-up).[11]”

In Materials and Methods, please explain why participants were requested to collect urine samples.

Authors: As sugested, we have removed this comment because it is not relevant for this analysis.

In Materials and Methods, it is not clear whether the dietary interview is assessed by a trained staff or the diet are self-reported, as reported in the limitations.

Authors: Dietary information was obtained both, by a trained staff and it was self-reported. The trained staff conducted an in-depth interview that lasted on average 50 minutes, but it was the participant himself who referred his own dietary intake (Reference # 13).

In the Result section, participants in the highest quartile of EI% at lunch have higher adherence to a Mediterranean diet. Please, comment this results in the Discussion section.

Authors: There is evidence in the literature of a protective relationship between adherence to the Mediterranean diet and MetS development (Rumawas ME, Meigs JB, Dwyer JT, McKeown NM, Jacques PF. Mediterranean-style dietary pattern, reduced risk of metabolic syndrome traits, and incidence in the Framingham Offspring Cohort. Am J Clin Nutr. 2009 Dec;90(6):1608-14). A greater adherence is associated with a lower incidence of the metabolic syndrome. In our analysis, the adjustment for the Mediterranean diet possibly raises the estimator from 1.62 to 1.71 as you can see in table 3. Despite the adjustment, the estimator for energy intake at lunch was far from reaching statistical significance 1.71 (0.73 to 3.97), so we don’t have enough evidence to say that a higher energy intake at lunch was associated with a higher incidence of the MetS.

In Discussion section, the putative role of insulin or melatonin as regulatory signals to peripheral circadian clocks remains largely speculative. Please could the Authors better explain the role of these hormones, if any, in influencing their present data?

Authors: Insulin and melatonin were not measured in this cohort. Therefore, our sole purpose was to indicate some possible mechanisms. We have simplified and shortened this paragraph.

The Authors state that the main strength of this study is its longitudinal design. However, the number of participants lost is very high (67%). Please, discuss this point in the limitation.

Authors: The reviewer is right in this point. Of note that 676 participants were excluded for having prevalent MetS. We acknowledge this point in the text . We have adding the following (line 323):

“Finally, we cannot rule out some selection bias due to the exclusion of participants with missing data.”

Again, among limitations, the Authors reported that the time of the meals was not collected.  This is of particular relevance in limiting the results of this study considering the emerging evidence on the positive impact of time-restricted feeding on metabolic outcomes (see also Currenti W et al Nutrients. 2021). Please, discuss this point. In the same context, no information is given on sleep preferences (see also Charlot A et al Nutrients. 2021). Again, comment this limitation.

Author: : The reviewer is right in this point, and they were added as limitations (line 323):

“On the other hand, the specific time of the six eating occasions was not recorded, although these are quite characteristic of the dietary habits in the Spanish population. Additionally, the influence of sleep preferences was not considered in the analyses.”

Reviewer 2 Report

Comments:

The study entitled as “A higher intake of energy at dinner is associated with incident 2 metabolic syndrome: a prospective cohort study in older adults” (Manuscript ID# nutrients-1353118)” n. indicating that higher energy intake (EI) at dinner is associated with a higher risk of metabolic syndrome (MetS) in older adults. Out of 607 participants, 101 developed MetS. Authors proposing that reducing EI at dinner might be a simple strategy to prevent MetS. The manuscript is well written and study looks interesting. There are few minor issues that should be addressed to enhance the outcome of the study:

Comment 1: Does higher EI intake at dinner is associated with MetS in younger population or in mid aged persons?

Comment 2:  Is there any difference in the development of MetS in males and females?

Comment 3:  Does physical activity reduces the risk of development of MetS? If yes, is it similar for males and females?

Comment 4:  High fat/high carbohydrate diet (western diet- fast foods and high energy drinks/soda/soft drinks) is linked with development of MetS. These type of food at lunch/dinner may lead to obesity. What is Authors’ opinion and findings?

Comment 5:  Does Day and night sleeping time affect the obesity status?  

Comment 6:  During Hyperglucemia/diabetes measurement in participants, only fasting glucose level was measured or insulin level is also analyzed? It will give in idea that low level of insulin in the participants with MetS is found or insulin resistance is developed.

Author Response

The study entitled as “A higher intake of energy at dinner is associated with incident metabolic syndrome: a prospective cohort study in older adults” (Manuscript ID# nutrients-1353118)” n. indicating that higher energy intake (EI) at dinner is associated with a higher risk of metabolic syndrome (MetS) in older adults. Out of 607 participants, 101 developed MetS. Authors proposing that reducing EI at dinner might be a simple strategy to prevent MetS. The manuscript is well written and study looks interesting. There are few minor issues that should be addressed to enhance the outcome of the study:

Comment 1: Does higher EI intake at dinner is associated with MetS in younger population or in mid aged persons?

Authors: Thank you for your comment. Unfortunately, in this cohort only older adults over 60 were followed up. To establish the presence of incident metabolic syndrome (MetS), laboratory data are needed, therefore, with our data we could not answer the question addressed by the reviewer. Previous cross-sectional studies conducted in Japan and Korea indicate that this association may be also be true among younger people, but there are no conclusive data from longitudinal studies.

Comment 2:  Is there any difference in the development of MetS in males and females?

Authors: As we previously published, the prevalence of the MetS was slightly higher in men before the age of 65, and slightly higher in women after the age of 65 (Guallar-Castillón et al. Magnitude and management of metabolic syndrome in Spain in 2008-2010: the ENRICA study. Rev Esp Cardiol . 2014 May;67(5):367-73.  doi: 10.1016/j.rec.2013.08.014). In the present analysis, after excluding those who already had prevalent MetS, the proportion of men and women was very balanced (51% of women). Among the 101 participants who developed metabolic syndrome during follow-up, 50 were men and 51 were women. Therefore, no relevant sex differences were observed in the development of incident MetS in this cohort of older adults. Analyses by sex were not presented due to lack of statistical power (50 events in 4 quartiles). In addition, to prevent confounding by sex, analyses were performed in sex-specific quartiles and analyses were adjusted for sex.

To make the manuscript clearer, we have added the number of men a women who developed MetS during follow-up (line 161).

Comment 3:  Does physical activity reduces the risk of development of MetS? If yes, is it similar for males and females?

Authors: For the main analyses, no significant differences were observed for physical activity during leisure time, or physical activity in the household (see table 2). Therefore, the analyses were not modified according to physical activity. Again, lack of statistical power prevents us from showing results stratified by physical activity. However, the main analyses in this manuscript were adjusted for physical activity during leisura time, in the household as well as the time spent watching TV, so we think that the influence of physical activity was correctly addressed.

Comment 4:  High fat/high carbohydrate diet (western diet- fast foods and high energy drinks/soda/soft drinks) is linked with development of MetS. These type of food at lunch/dinner may lead to obesity. What is Authors’ opinion and findings?

Authors: The consumption of ultra-processed foods has been associated with the development of MetS, while at the same time, ultra-processed foods are energy-dense (and slow in producing satiety). Another way to address the present question is whether the percentage of energy from ultra-processed foods consumed at dinner is associated with increased incidence of MetS. This association is plausible, however, it is beyond the scope of this manuscript.

Comment 5:  Does Day and night sleeping time affect the obesity status? 

Authors: In the main analyses, the distribution of total sleeping time were similar between extreme quartiles 7.2 hours vs 7.2 hours (results considered for dinner in table 2). When we considered the sleepint time during the night (witout considering the siesta time) results were also similar between extreme quartiles (6.8 hours vs 6.9 hours). So, although this distinction could theoretically be important, it was not relevant in our cohort.

Comment 6:  During Hyperglucemia/diabetes measurement in participants, only fasting glucose level was measured or insulin level is also analyzed? It will give in idea that low level of insulin in the participants with MetS is found or insulin resistance is developed.

Authors: To define MetS the harmonised definition was followed, and only fasting blood glucose was considered. We don’t have data on insulin at the end of follow-up.